# Intratracheal Administration of Stem Cell Membrane-Cloaked Naringin-Loaded Biomimetic Nanoparticles Promotes Resolution of Acute Lung Injury

**DOI:** 10.3390/antiox13030282

**Published:** 2024-02-26

**Authors:** Hua Jin, Yue Zhao, Yinlian Yao, Shilong Fan, Renxing Luo, Xin Shen, Yanyan Wang, Jiang Pi, Gonghua Huang

**Affiliations:** 1Guangdong Provincial Key Laboratory of Medical Molecular Diagnostics, The First Dongguan Affiliated Hospital, Guangdong Medical University, Dongguan 523808, China; luorenxing2020@163.com (R.L.); yanyan.wang@gdmu.edu.cn (Y.W.); 2School of Pharmacy, Guangdong Medical University, Dongguan 523808, China; yuezhao0520@163.com (Y.Z.); yinlian_yao@163.com (Y.Y.); fsl2000123@163.com (S.F.); 15515792918@163.com (X.S.); 3School of Medical Technology, Guangdong Medical University, Dongguan 523808, China; jiangpi@gdmu.edu.cn

**Keywords:** naringin, acute lung injury, stem cell membrane, biomimetic nanoparticle, intratracheal instillation, macrophage polarization

## Abstract

Cytokine storm and ROS overproduction in the lung always lead to acute lung injury (ALI) and acute respiratory distress syndrome (ARDS) in a very short time. Effectively controlling cytokine storm release syndrome (CRS) and scavenging ROS are key to the prevention and treatment of ALI/ARDS. In this work, the naringin nanoparticles (Nar-NPs) were prepared by the emulsification and evaporation method; then, the mesenchymal stem cell membranes (CMs) were extracted and coated onto the surface of the Nar-NPs through the hand extrusion method to obtain the biomimetic CM@Nar-NPs. In vitro, the CM@Nar-NPs showed good dispersity, excellent biocompatibility, and biosafety. At the cellular level, the CM@Nar-NPs had excellent abilities to target inflamed macrophages and the capacity to scavenge ROS. In vivo imaging demonstrated that the CM@Nar-NPs could target and accumulate in the inflammatory lungs. In an ALI mouse model, *intratracheal* (*i.t.*) instillation of the CM@Nar-NPs significantly decreased the ROS level, inhibited the proinflammatory cytokines, and remarkably promoted the survival rate. Additionally, the CM@Nar-NPs increased the expression of M2 marker (CD206), and decreased the expression of M1 marker (F4/80) in septic mice, suggesting that the Nar-modulated macrophages polarized towards the M2 subtype. Collectively, this work proves that a mesenchymal stem cell membrane-based biomimetic nanoparticle delivery system could efficiently target lung inflammation via *i.t.* administration; the released payload inhibited the production of inflammatory cytokines and ROS, and the Nar-modulated macrophages polarized towards the M2 phenotype which might contribute to their anti-inflammation effects. This nano-system provides an excellent pneumonia-treated platform with satisfactory biosafety and has great potential to effectively deliver herbal medicine.

## 1. Introduction

Cytokine storm is an overactivation of the immune system caused by infections, drugs, etc., which can occur rapidly and lead to organ failure, life-threatening conditions, and even death [1]. In the COVID-19 epidemic, most patients with COVID-19 developed acute respiratory distress syndrome (ARDS) [2,3,4], multiple organ failure [5], and even death. It is noticeable that one of the most important mechanisms underlying the deterioration of this disease is cytokine storm release syndrome (CRS). CRS occurs in the pulmonary airways and develops into severe pneumonia, such as ALI (acute lung injury) and/or ARDS [6]. Additionally, reactive oxygen species (ROS) is closely associated with organ damage in ALI [7]. The overproduction of ROS could induce the activation of lung macrophages and infiltration of neutrophils, and simultaneously invade and spread into lung epithelial and endothelial cells, which promptly leads to tissue damage and organ dysfunction [8]. Thus, effectively controlling the CRS and scavenging the ROS are key strategies for treating severe pneumonia and other serious infectious diseases.

Currently, glucocorticoids, antibiotics, neutralizing antibodies (such as anti-IL-6), and stem cell-based therapies are mainly employed in clinics to calm the CRS and treat CRS-induced pneumonia [9,10,11]. Although these therapeutics exert highly effective anti-inflammatory merits, the off-targeting effect determines their different degrees of systemic immunosuppressive side effects [12]. Currently, there is no specific treatment for ALI/ARDS. In clinics, the main supportive measures include protective lung ventilation and restrictive fluid resuscitation, among others. Although supportive treatment can alleviate patients’ clinical symptoms, it does not significantly improve their prognosis. The mortality rate of sepsis-induced ALI/ARDS remains as high as 46.1% [13].

Therefore, finding new targets and developing targeted strategies are of great significance for controlling clinical pneumonia from mild to severe, reducing severe mortality, shortening hospitalization time, and improving the life quality of patients. Plants are the most abundant organisms on Earth in terms of biomass. Compounds and natural products derived from plants have been widely used for the treatment of various diseases. Naringin (Nar), a natural bioactive flavanone, which is rich in grapefruits, shows fabulous antioxidant and anti-inflammatory properties. Previous studies indicated that Nar could attenuate airway inflammation and ameliorate lung dysfunction [14], while the poor solubility and off-targeting of Nar greatly limit its therapeutic efficiency. Thus, targeted delivery of Nar into the inflammation lesions is a key strategy to increase its anti-inflammation efficacy.

Biodegradable nanoparticle-based delivery systems have been emerging as attractive pharmacological vehicles to improve bioavailability and biocompatibility of drugs due to their enhanced water-solubility and sustained and controlled release properties [15]. However, nanoparticle capture and elimination by the immune system are great obstacles to drug delivery. Currently, the cloaked nanoparticles with cell membranes represent a promising strategy to evade phagocytosis and avoid immunogenicity by the immune system. This kind of biomimetic nanoparticles simultaneously possesses the functions of drugs loaded in nanoparticles and the biological functionalities of the source cells inherited from the cell membranes. As expected, the cell membrane-cloaked nanoparticle-delivery system could evade immune elimination, prolong circulation time, and even target to disease regions by the nature of the homing ability of the cells [16]. In the past decades, stem cells have been studied and utilized for repair and regenerative medicine [17]. The inherent properties of stem cells determine that the stem cell-based biomimetic nanoparticles can be effectively targeted to tumor or inflammatory sites, which provide sustained circulation and boosted drug accumulation at target sites, augmenting therapeutic efficacy and safety [18]. Additionally, stem cells can be obtained from various sources, including bone marrow, amniotic cells, adipose tissue, umbilical cord, and placental tissue. Stem cell membrane functionalized nanoparticles combine anti-inflammatory and antimicrobial drugs indicating excellent therapeutic effects for sepsis treatment [19].

Traditional anti-inflammation drugs are mainly administered orally or by injection, but these administration methods always bring issues of adverse reactions in the liver and gastrointestinal tract, as well as the development of drug resistance with long-term use. In pulmonary diseases, the lung is the main targeted organ to achieve drug local delivery. Thus, delivering drugs directly to the target site through the bronchial-pulmonary route can not only enhance drug efficacy but also reduce the occurrence of adverse reactions and drug resistance. Researchers comprehensively compared the therapeutic activity, biodistribution, and lung cell targeting characteristics of three anti-inflammatory drug delivery routes—*intratracheal* (*i.t.*), *intravenous* (*i.v.*), and *intraperitoneal* (*i.p.*) [20]. The results indicated that *i.t.* administration of NPs exhibited excellent efficacy in the ALI treatment compared with other administrated methods.

Although most Chinese herbal medicines, such as Nar, exhibit potent anti-inflammatory and antioxidant activities at the cellular level, their clinical efficacy and application are greatly limited due to their poor water solubility, low bioavailability, off-target effects, and difficulties in cellular uptake. Thus, in this work, to overcome these disadvantages, stem cell membrane-cloaked PLGA NPs were synthesized and delivered Nar to the inflammatory lesions in vivo. We successfully constructed a herbal ingredient (Nar) loading PLGA NPs for ALI management. The Nar-NPs were further modified with LPS-treated bone marrow derived-mesenchymal stem cell (BMSC) membrane, to exert targeting delivery to the inflammatory microenvironments. The biocompatibility, inflammation-targeting, antioxidant capacity, and anti-inflammatory efficiency were determined both in vitro and in vivo. The as-synthesized biomimetic platform (CM@Nar-NPs) showed highly efficient targeting and Nar delivery to the macrophages and alveolar epithelial cells as demonstrated by fluorescent imaging and flow cytometry analyses. The lung inflammation-targeting and accumulation of CM@Nar-NPs were also confirmed by IVIS imaging in septic mice. In the LPS-induced ALI mouse model, the CM@Nar-NPs showed efficient therapeutic efficacy, as evidenced by improved survival rate, decreased pulmonary permeability and cytokine release, and organ damage protection. Additionally, the CM@Nar-NPs exhibited excellent antioxidant capacity in the inflamed RAW264.7 cells and in septic mice. To further determine the protective mechanism of CM@Nar-NPs, infiltrated macrophages in the lung were isolated and analyzed. We propose that this kind of biomimetic nano-platform might be an efficient strategy to deliver drugs for counteracting infectious diseases, such as pneumonia.

## 2. Materials and Methods

### 2.1. Animals in the Study

Eight–twelve-week-old female C57BL/6J mice were purchased from the SPF (Beijing) Biotechnology Co., Ltd., Beijing, China. Experimental procedures using mice in this work were reviewed and approved by the ethical review board of Guangdong Medical University, and all the experiments were performed following relevant guidelines and regulations of the Animal Ethics Committee of Guangdong Province, China.

### 2.2. Preparation of Nar-NPs

The Nar-NPs were synthesized using the emulsification and evaporation method as previously described [21] with a little modification. Briefly, 20 mg Naringin (Shanghai Yuanye Bio-Technology Co., Ltd., Shanghai, China) and 60 mg PLGA-PEG (lactide/glycolide 50:50, Sigma, Ronkonkoma, NY, USA) were co-dissolved in 5 mL of dichloromethane (Tianjin Damao Chemical Reagent Factory, Tianjin, China) as the oil phase (O); 20 mL of PVA (1%, *w*/*w*, Sigma) was used as the external water phase (W). Firstly, the O phase was ultrasonic for 40 s in the ice bath to form the first emulsification. Then, the first emulsification was dropped in the W phase and ultrasonicated for another 40 s to form the second emulsification. After that, the second emulsification was added to 100 mL water and stirred for 6 h for organic regent evaporation and nanoparticle hardening. Finally, the nanoparticles were harvested by centrifuging at 12,000 rpm for 20 min and washed 3 times using ultrapure water. The harvested NPs were stored at 4 °C or lyophilized for long-term storage.

### 2.3. Preparation of Cell Membrane Cloaked CM@Nar-NPs

Bone marrow-derived mesenchymal stem cells were isolated according to previous published protocols [22]. The stem cells were then treated with 1 µg/mL LPS for 24 h. The harvested cells were washed three times with PBS. The extraction of cell membranes followed Shi’s protocol [23]; briefly, 5 × 10^7^ cells were collected and suspended in 3 mL of hypotonic lysing buffer containing 50 mM Tris, 150 mM NaCl, 1% NP-40, 0.25% sodium deoxycholate, and 10% (*v*/*v*%) benzylsulfonyl fluoride. Then, the cells were kept in an ice bath for 15–30 min, followed by 3 rounds of freeze–thaw processes (frozen at −80 °C and thawed at 37 °C). The cell suspension was then centrifuged at 800× *g* for 15 min. The supernatant was collected and further centrifugated at 18,000× *g*, 4 °C for 60 min. The final precipitate was collected as cell membranes, and the protein content was analyzed by bicinchoninic acid protein assay. Finally, the mixture was extruded via a Hand Extruder to obtain CM@Nar-NPs.

### 2.4. Characterization of NPs

The size distribution and zeta potential of the nanoparticles were assayed using a Nano Particle Analyzer SZ-100 (Horiba Scientific, Austin, TX, USA). The morphology of NPs was visualized by scanning electron microscopy (SEM, Philips Co., Eindhoven, The Netherlands).

Determination of Nar encapsulation rate was as follows. The free Nar, Nar-NPs, and CM@Nar-NPs were full wavelength scanned using a UV–vis spectrum (UV 6000), and the data showed that the UV absorption peak of free Nar and Nar-NPs was at 285 nm and 282 nm, respectively. The absorption peak shift was not significant. It was supposed that Nar was successfully encapsulated into the NPs.

To measure the drug (Nar) loading rate of Nar-NPs, 10 mg lyophilized nanoparticles were dissolved in 1 mL of methanol, and then the amount of Nar in solution was determined by high-pressure liquid chromatography (HPLC). HPLC detection was performed using a C18 column (5 µm, 250 mm × 4.6 mm); whereas, the mobile phase, consisting of methanol and 0.1% acetic acid (88:12) (*v*/*v*), was maintained at a flow rate of 1.0 mL/minute. The ultraviolet detector wavelength was 285 nm and the injection volume was 20 µL. The loading of Nar was calculated the concentration of Naringin based on the standard curve of Nar at 285 nm.

### 2.5. Hemolysis Assay

To determine the in vivo biosafety of the compounds, hemolysis of the red blood cells treated with the designated compounds was determined [24]. Whole blood was drawn from healthy C57BL/6J mice and anticoagulated using heparin, and then centrifuged at 3500 rpm for 10 min. The erythrocytes were isolated from the sediments and washed with PBS 3 times. A 4% erythrocytes suspension (*v*/*v*, in PBS) was blended with different formulations of Nar. The erythrocyte sample lysed using pure water was used as a positive control (hemolysis group), and the erythrocyte sample diluted using PBS was used as a normal control group. After incubating with 100 μg/mL of compounds at 37 °C for 4 h, all the samples were centrifuged at 3500 rpm, 4 °C, for 5 min to collect the supernatant and measure its absorbance value at 550 nm by a microplate reader. Also, the red blood cells were added to a 12-well plate and imaged using a living cell imaging system (EVOSFL Auto, Invitrogen, Carlsbad, CA, USA).

### 2.6. SDS-PAGE Analysis of Retention Proteins from Cell Membranes and NPs

The membrane proteins from stem cell membrane vesicles or CM@Nar-NPs were extracted following the instructions of the Membrane and Cytosol Protein Extraction Kit P0033 (Beyotime, Shanghai, China). A BCA protein assay kit (Solarbio, Beijing, China) was used to determine the protein concentration of isolated CM. Stem cell membrane protein samples and CM@Nar-NPs were mixed with loading buffer (Beyotime, China) and denatured at 95 °C. Protein samples of 30 μg were separated by 10% SDS-PAGE gel, and then, the gel was stained with Coomassie Blue for 2 h and decolored overnight for imaging.

### 2.7. In Vitro Targeting and Biodistribution of the NPs

Due to the weak autofluorescence of Nar, Cumarin 6 (C6) was co-loaded into the Nar-NPs as a fluorescent marker. RAW264.7 cells and human umbilical vein endothelial cells (HUVECs) were provided by the Cell Library of Guangdong Provincial Key Laboratory of Medical Molecular Diagnostics, Guangdong Medical University. The cells were cultured in DMEM (Gibco, Grand Island, NY, USA) with 10% fetal bovine serum (Gibco) containing 100 µg/mL streptomycin and 100 IU/mL penicillin at 5% CO_2_ and 37 °C. The cells were cultured with C6/Nar-NPs for 4 h, and then washed twice with PBS to remove unbound nanoparticles followed by trypsinization. The fluorescent images were obtained using a living cell imaging system (EVOSFL Auto, Invitrogen, USA). Also, flow cytometry analysis [25] was performed at the FITC channel. Results were expressed in terms of the mean fluorescence intensity (MFI) of C6 in the cells.

For further intracellular location determination, the cells were cultured with NPs for 2 h, then, the lyso- or mito-tracker was added to the culture for another 2 h, rinsed twice with PBS, and stained with DAPI (to locate the nucleus); the cells were then sealed with glycerol before being imaged under a Leica SP8 confocal microscope (Leica Microsystems Inc., Wetzlar, Germany).

### 2.8. CCK8 to Assay Cell Viability

To examine the effects of Nar or Nar-NPs on RAW264.7 cells and HUVEC cells, the CCK8 assay was carried out to assess the cell viability and proliferation. In brief, RAW264.7 cells or HUVEC cells were plated in a 96-well plate at a density of 1 × 10^4^ cells/well. Cells were treated with different concentrations of Nar, Nar-NPs, or CM@Nar-NPs for 24 h. Then, a CCK8 assay was performed according to the manufactures’ instructions.

### 2.9. In Vitro ROS Level Assay

The RAW264.7 cells suspension was seeded into 6-well plates (10^5^ cells/well) and cultivated for 24 h. Next, different formulations of Nar were added to the cells for 24 h. The cells were harvested to incubate with 10 nM of DCFH-DA in DMEM medium for 30 min in the dark at 37 °C and washed twice with PBS. The MFI (mean fluorescence intensity) of DCF-HA in the cells was measured by flow cytometer using the FITC channel.

### 2.10. In Vivo Targeting and Cellular Uptake of NPs

For in vivo imaging, indocyanine green (ICG) was co-loaded into the NPs as fluorescent markers. LPS-induced septic mice were *i.t.* administered with 50 μL of Nar/ICG-NPs or CM@Nar/ICG-NPs. For semiquantitative analysis, the MFI of mice were collected 4 h after *i.t.* administration with NPs and imaged using an in vivo imaging system.

To further analyze the NPs endothelia targeting, the ALI mice were *i.t.* administrated with coumarin 6 (C6)-loaded NPs (5 mice/group); after 4 h, the lungs were collected and isolated cells used to determine the uptake of NPs using flow cytometry. In addition, healthy mice without LPS treatment were also *i.t.* administered with CM@Nar/C6-NPs, which served as healthy control group.

Also, lung tissues were isolated and rinsed thoroughly with PBS and embedded in optimal cutting temperature compound (Sukura, Tokyo, Japan). Sections (5~10 μm) were stained with DAPI to locate the nuclei. Images were acquired using a Leica SP8 confocal microscope.

### 2.11. LPS-Induced ALI Mouse Model and Experimental Groups

The mice were randomly divided into four groups, and *i.t.* administered with Nar (10 mg/kg), Nar-NPs (10 mg/kg), CM@Nar-NPs (10 mg/kg), or PBS, separately. Three hours later, mice were *i.p.* injected with LPS (Beijing Solarbio Science & Technology Co., Ltd., Beijing, China) at 7.5 mg/kg in 50 μL PBS. After 24 h, mice were sacrificed and samples (blood, bronchoalveolar lavage fluid, lung tissues, etc.) were collected.

### 2.12. Assay of ROS Level in Lungs

The mice were anesthetized and systemic blood was perfused before collecting tissues. The lungs and livers were removed from the mice and immediately washed with PBS, and then submersed in DMEM medium with 100 μM DCFH-DA for 30 min at 37 °C in the dark. After washing twice with DMEM medium, the fluorescence intensity of tissues was imaged using a Small Animal Live Imaging System (Eastman Kodak Co., Rochester, NY, USA).

### 2.13. Measurement of Lung Vascular Leakage

Evans blue-conjugated albumin (EBA) extravasation assay to assess vascular permeability was performed according to the instructions of previous studies [26]. Briefly, EBA (20 mg/kg) was injected retro-orbitally 45 min before sacrificing mice. Lungs were perfused with PBS to remove blood. Lung homogenates were incubated with 2 volumes of formalin for 18 h at 60 °C, centrifuged at 5000× *g* for 30 min, and the optical density (OD) of the supernatant was determined spectrophotometrically at 620 nm. The extravasated EBA in lung homogenate was expressed as μg of EBA per g of lung tissue.

### 2.14. Survival Rate

After a lethal dose of LPS (10 mg/kg) challenge, mice were given normal access to water and food, and monitored four times a day for 7 days. Moribund mice were identified and euthanatized using CO_2_ followed by cervical dislocation. On day 7, all the surviving mice were euthanatized.

### 2.15. Cytokine Assay

The whole blood with EDTA originating from septic mice was centrifuged at 3000 rpm for 10 min at 4 °C, and the plasma was isolated for ELISA assay of IL-6 and IL-1β levels according to the protocols provided by the manufacturer (Thermo Fisher Scientific, Waltham, MA, USA).

### 2.16. Histology

Lung tissues were fixed by 5 min of instillation of 10% formalin through trachea catheterization at a trans-pulmonary pressure of 15 cm H_2_O, and then fixed in 10% formalin at room temperature for 48 h. Then, the lungs were embedded in paraffin, cut into 5 µm sections, and stained with hematoxylin and eosin (H&E).

### 2.17. Lung Mononuclear Cells Isolation

Lung mononuclear cells were prepared as previously described [27]. Briefly, lung tissues were sliced into small pieces and incubated at 37 °C for 45 min with collagenase IV (1 mg/mL; Life Technologies, Carlsbad, CA, USA) in RPMI-1640 medium (HyClone, Logan, UT, USA) supplemented with 5% fetal bovine serum (FBS; HyClone); cells were isolated by gradient centrifugation over 38% Percoll (GE Healthcare Life Sciences, Marlborough, MA, USA). After erythrocyte lysis with ACK lysis buffer (Gibco), the cells were harvested for analyses.

### 2.18. Flow Cytometric Analysis for Macrophage Polarization in Lung

For surface staining, lung mononuclear cells were labeled with FITC-mouse CD11b (48-0112-82, eBioscience, San Diego, CA, USA), PE/cyanine7- anti-mouse CD11c (117317; Biolegend, San Diego, CA, USA), APC-anti-mouse F4/80 (123115; Biolegend), FITC-anti-mouse CD206 (141703; Biolegend), and the matching control isotype IgG (MCA421; AbD Serotec, Oxfordshire, UK) in FACS buffer (PBS with 2% FBS) for 30 min at 4 °C. M1 macrophages were identified as F4/80+CD11c+CD206− and M2 macrophages were identified as F4/80+CD11c−CD206+ cells. Flow cytometry data were acquired on BD LSRFortessa X-20 (BD Biosciences, Franklin Lakes, NJ, USA) and analyzed using FlowJo^TM^ software v10 (Tree Star, Ashland, OR, USA).

### 2.19. Biosafety Study of Different Formulations of Nar

To determine the biosafety in vivo, different formulations of Nar (Free Nar, Nar-NPs and CM@Nar-NPs) were *i.t.* administered to normal healthy mice (dose: 10 mg/kg body weight, 5 mice/group) for one week (once/day). On the 8th day, the mice were euthanized and the main organs were harvested for tests. The pathological sections of the main organs (spleen, liver, and kidney) were stained with hematoxylin and eosin (H&E).

### 2.20. Statistical Analysis

Results are expressed as mean ± SEM. Statistical significance was determined by one-way ANOVA with a Games–Howell post hoc analysis for multiple-group comparisons. Two-group comparisons were analyzed by the two-tailed unpaired Student’s *t*-test.

## 3. Results

### 3.1. Characterization of CM@Nar-NPs

A classical emulsification and evaporation method was used to prepare the PLGA NPs (Figure 1A). During the synthesis process, Nar was added into the oil phase of the PLGA NPs. The harvested Nar-NPs were mixed with stem cell membranes and extruded through a polycarbonate porous membrane to form CM@Nar-NPs (Figure 1B). This as-synthesized biomimetic nano-system was supposed to achieve the inflammation targeting and facilitating the delivery of Nar into the lungs in LPS-induced septic mice (Figure 1C).

A Nano Particle Analyzer SZ-100 (Horiba Scientific) and scanning electron microscope (SEM) were employed to characterize the size and morphology of Nar-NPs and CM@Nar-NPs. As shown in Figure 2A, the sizes of Nar-NPs were about 400 nm, and these were increased slightly to ~470 nm with the CM coating. Notably, the zeta potential of the Nar-NPs was about ~10 mV, indicating that the Nar-NPs were easily gathered together, while after coating with CM, the zeta potential of the CM@Nar-NPs was decreased to ~18 mV, implying a good dispersion of NPs (Figure 2B). The SEM image (Figure 2C) showed that both kinds of NPs presented a spherical and uniform morphology with a diameter of ~400 nm. The SEM results further confirmed the good dispersion of the CM@Nar-NPs, which was consistent with the results of Zeta potential.

By FTIR spectroscopy analysis (Figure 2D), the Nar-loaded PLGA-PEG NPs showed specific FTIR absorption peaks at 2948.1 cm^−1^ of PLGA-PEG, and 1104.8 cm^−1^ of Nar, respectively, confirming the successful encapsulation of Nar into the Nar-NPs.

The UV–vis spectrum of the NPs was measured to determine the loading of Nar in the NPs. As shown in Figure 2E, the absorption spectrum of Nar exhibited the characteristic absorption peaks at 285 nm. The absorption peak of Nar-NPs and CM@Nar-NPs was 282 nm, which was closely consistent with that of free Nar, indicating a successful encapsulation of Nar into the NPs. The loading capacity of Nar was 78% in the Nar-NPs and CM@Nar-NPs, which was calculated based on the standard curve of Nar at their UV absorption peak at 285 nm.

To further confirm the CM coating onto the Nar-NPs, the protein ingredients of the CM and CM@Nar-NPs were analyzed by SDS-gel electrophoresis [28]. As shown in Figure 2F, the CM@Nar-NPs contained characteristic proteins preserved by stem cell membranes (indicated by a red circle). Thus, these results indicated that the proteins of the CM were successfully coated or conjugated on the surface of the Nar-NPs.

The release kinetics of Nar in vitro are shown in Figure 2G. The initial release of free Nar appeared to be greater than those from the Nar-NPs and CM@Nar-NPs in the first 2 h. At 8 h, the free Nar almost released more than 80%, while the nanoparticles loaded with Nar only released ~30%. The data suggested that the nano-particle platform could significantly temper the initial release of Nar and prolong the action of Nar in the body.

### 3.2. In Vitro Biosafety of Different Formulations of Nar

Given that the biocompatibility and biosafety of nanoparticle-based systems have been known as the most important factors for their clinical application, here we employed RAW264.7 cells and HUVECs to determine the cytotoxicity of Nar. As shown in Figure 3A, the free Nar slightly inhibited the cell viability of macrophages; less than 80 μg/mL of Nar did not induce cytotoxicity on macrophages, while 160 μg/mL of Nar induced significant toxic effects on macrophages. Interestingly, low concentrations (≤80 μg/mL) of the CM@Nar-NPs showed minor toxic effects, even slightly increasing the proliferation of HUVEC cells, implying that the CM@Nar-NPs had good biocompatibility with the lung epithelial cells (Figure 3B). Additionally, 160 μg/mL of Nar exhibited significant inhibition or killing effects on both the RAW264.7 cells and HUVECs, indicating that a high dose of Nar could induce cytotoxic effects. Interestingly, 160 μg/mL of the CM@Nar-NPs did not induce significant cytotoxic effects on both kinds of cells.

To further confirm the biosafety of the Nar-NPs and CM@Nar-NPs in vivo, the hemolytic properties were evaluated using mouse red blood cells. Erythrocytes lysed with water were used as the hemolysis-positive group and its hemolysis rate was set at a value of 1.0. The free Nar, Nar-NPs, and CM@Nar-NPs were used at the same concentration of 80 μg/mL (calculated based on the Nar loading into the NPs). As shown in Figure 3C–E, the free Nar showed slight hemolysis. Interestingly, after being formulated into the NPs, erythrocytes treated with the Nar-NPs or CM@Nar-NPs did not show obvious hemolysis. In addition, the shapes of the erythrocytes with different treatments were also imaged under light microscope. The same number of erythrocytes was added into a 12-well plate, and the same concentration of different formulations of Nar was added and cultured for 30 min at 37 °C; pure water or PBS was also added into the wells as positive or negative controls, respectively. As shown in Figure 3E, the water-treated erythrocytes were almost completely destructed, and only a few cell fragments were left, while the PBS-treated group showed amounts of normal, healthy erythrocytes with intact double concave disc shapes. Notably, the erythrocytes represented slightly decreased-in-number and deformed shapes, implying that free Nar might induce slight hemolysis. Interestingly, erythrocytes co-cultured with the Nar-NPs and CM@Nar-NPs represented intact membranes and healthy shapes, suggesting that the nanoparticle-based platform could significantly decrease hemolysis and improve the biocompatibility of drugs in vivo. Collectively, these data indicated that the CM@Nar-NPs could significantly decrease the cytotoxicity and improve the biosafety of Nar.

### 3.3. In Vitro Targeting and Biodistribution of NPs

The number of macrophages increased significantly in the ALI lungs, suggesting that macrophages were involved in the occurrence and development of ALI/ARDS [29]. Thus, it is very important to target delivery of drugs to the macrophages. Due to the weak fluorescence of Nar, coumarin 6 (C6) was also co-encapsulated into the NPs to be the fluorescent marker. To determine the in vitro inflammation targeting of the CM@Nar-NPs, RAW264.7 cells were stimulated with LPS to induce the inflammation condition. After 2 h of co-culture, the NPs could successfully enter the cells, which was confirmed by the fluorescent imaging (Figure 4A,B) and the MFI in the cells assayed by flow cytometry (Figure 4C,D). Interestingly, the cellular uptake of NPs in LPS-stimulated cells was much more than that of the rest of the RAW264.7 cells without LPS stimulation, indicating that the CM@C6/Nar-NPs tended to target to the inflamed cells.

To further determine the subcellular location of the CM@C6/Nar-NPs, the lysosomes and mitochondria were stained with lyso-tracker and mito-tracker, respectively; the confocal images showed that the CM@C6/Nar-NPs fluorescence was partly colocalized with the lyso-tracker and mito-tracker, indicating that the CM@C6/Nar-NPs were internalized and entered into the lysosomes and mitochondria of macrophages. The results indicated that the CM@NPs system could successfully deliver drugs into the mitochondria of macrophages in inflamed lungs, which guaranteed their treatment effects on ALI.

### 3.4. In Vivo Targeting and Accumulation of NPs in Inflamed Lungs

To further confirm the targeting and accumulation of the CM@Nar-NPs in inflammatory lungs, several organs of ALI mice were imaged using an IVIS imaging system. Indocyanine green (ICG) was co-loaded into the nanoparticles as the fluorescent marker. Figure 5A indicates the protocol and time axis of the in vivo targeting assays. First, mice were *i.p.* injected with LPS (7.5 mg/kg) to induce ALI pneumonia. Free Nar/ICG, Nar/ICG-NPs, or CM@Nar/ICG-NPs were then *i.t.* administered to visualize the tracks of NPs. Four hours after *i.t.* administration of NPs or free Nar, the lungs and other main organs were collected and imaged by in vivo imaging systems (IVISs). The accumulation of NPs in the lung is shown in Figure 5B. We observed that both kinds of Nar/ICG-NPs and free Nar/ICG showed strong fluorescence in the inflamed lungs 4 h post-*i.t.* administration. Notably, the Nar/ICG-NPs, particularly the CM@Nar/ICG-NPs, showed much stronger fluorescence intensity in the lungs. The lower fluorescence of free ICG/Nar might be associated with the fast clearing out by the immune system. The quantitative analysis of the IVIS demonstrated that the accumulation of CM@Nar/ICG-NPs in the lungs was five-fold higher and three-fold higher than that of free Nar/ICG and Nar/ICG-NPs, respectively (Figure 5C). The results suppose that stem cell membrane decoration significantly improves the targeting and accumulation of Nar-NPs in the inflamed lungs.

The flow cytometry results demonstrated that the CM coating significantly improved the delivery of NPs in the lung cells. The lung cell accumulation of CM@NPs was improved more than two-fold than that of the non-modified NPs (Figure 5C). The accumulation and cellular uptake of nanomedicine in the tissue lesions are prerequisites for their therapeutic efficacy. To further detect the special inflammation targeting of CM@NPs, we also assayed the in vivo cellular uptake of the CM@Nar/C6-NPs in the inflamed lungs and healthy lungs (Figure 5D,E). Notably, there was almost no CM@Nar/C6-NPs accumulation and retention in the cells (upper panel in Figure 5D,E) originating from healthy lungs. While there was a mass accumulation of the CM@Nar/C6-NPs in the lungs originated from ALI mice, the uptake of the CM@Nar/C6-NPs was much higher than that of the Nar/C6-NPs, demonstrating the enhanced targeting with the CM coating. To further confirm the cellular uptake of NPs in lung tissues, the frozen sections of ALI lungs were stained with DAPI to locate the nuclei and CD31 antibody to locate endothelial cells. Figure 5F demonstrated that the biomimetic CM@Nar/C6-NPs were more uptaken by lung cells comparing with non-modified Nar/C6-NPs. And, the co-location of NPs with CD31 antibody indicated that the NPs could be targeted delivered into endothelial cells, implying that this biomimetic nanoparticle system would be potential to repair the endothelia barrier.

Together, these in vivo data indicate that the *i.t.* delivery of CM-coated NPs can specifically improve the targeting of NPs to the inflamed lungs and cells. In healthy mice, there was almost no accumulation and retention of the CM@NPs in the lungs, implying the good biosafety of this kind of inflammation-targeting biomimetic NP in healthy mice.

### 3.5. CM@Nar-NPs Reduced Vascular Permeability of ALI Lung

The pathobiology of ALI includes a loss of alveolar–capillary membrane integrity, neutrophil infiltration, and release of pro-inflammatory cytokines. Among all these pathological processes, the increased lung vessel permeability plays a vital role in neutrophil infiltration and cytokine storm. Therefore, restoration of microvascular barrier function is essential for maintaining tissue fluid homeostasis and reversing lung edema. Lung vascular permeability was assessed by determination of Evans blue albumin (EBA) flux (permeability to protein) (Figure 6A) [28]. Mice were *i.t.* administered with different formulations of Nar, and 3 h later *i.p.* administered with LPS (7.5 mg/kg) to induce ALI pneumonia. At 23.5 h after the LPS challenge, mice were injected intravascularly with EBA, and after 30 min, the mice were euthanized to determine the EBA flux in the lungs. LPS treatment increased EBA flux in the lungs at 24 h post-LPS challenge, compared to basal controls (Figure 6B,C). There were no discernable improvements in the lung permeability in free Nar-treated mice compared to the PBS treatment. Mice treated with the Nar-NPs, especially the CM@Nar-NPs, showed remarkably decreased levels of EBA flux. It demonstrated that the therapeutics of the CM@Nar-NPs resulted from the targeted delivery of Nar to the inflamed lungs and inhibition of pneumonia permeability.

### 3.6. Survival Rate

The therapeutic effect of different formulations of Nar on ALI was determined using an LPS-induced ALI mouse model (Figure 7A). It is well known that LPS *i.p.* injection induces septic characteristics such as neutrophil margination and accumulation in the alveolar micro-vessels, which promptly can cause cytokine storm in the lungs, leading to organ dysfunction, even death [30]. To determine the survival improvement of the CM@Nar-NPs, a standard ALI mouse model was prepared by challenging with a lethal dose of LPS (10 mg/kg body weight). After the previous 3 h treatment with a lethal dose of LPS, the mice were *i.t.* administrated with 10 mg/kg of Nar, Nar-NPs or CM@Nar-NPs, and then the mice were kept under intensive observation. All the septic mice treated with PBS died within 24 h, indicating the ALI mouse models were successfully established. In the free Nar-treated group, only 20% of the mice could survive in the first 36 h. The Nar-NPs group mice had 60% survival. The CM@Nar-NPs treatment could largely protect the mice from LPS-induced ALI (Figure 7B).

### 3.7. In Vitro and In Vivo ROS Scavenging Activity

As ROS production may contribute to lung injury during pneumonia, the ability to scavenge ROS of Nar was determined using 2,7-dichloro-di-hydrofluorescein diacetate (DCFH-DA) staining in vitro and in vivo. The results showed that all the formulations of Nar significantly inhibited the generation of ROS in the LPS-challenged RAW264.7 cells. Among the three kinds of compounds, the CM@Nar-NPs exhibited the most obvious reduction in ROS production (Figure 8A). Further, we found that the CM@Nar-NPs treatment significantly inhibited the generation of ROS in the lungs and livers during ALI pneumonia. In vivo results showed that the free Nar treatment induced a limited ROS-scavenging capability compared with that of the Nar-NPs and CM@Nar-NPs (Figure 8C). Moreover, the quantitative statistical results in Figure 8B,D showed that all the formulations of Nar, even free Nar, had excellent ROS-scavenging capability. This supposed that the antioxidant activity and therapeutic efficacy of Nar were significantly improved through loading into the biomimetic nano-platform, especially in the lungs. Additionally, the Nar-NPs, especially the CM@Nar-NPs, had prominent ROS-scavenging capability.

### 3.8. Anti-Inflammation Efficacy Assay

To demonstrate the therapeutic efficacy and anti-inflammation activity by biomimetic targeted delivery, we *i.t.* administered the mice with PBS, free Nar (10 mg/kg), or Nar-NPs, CM@Nar-NPs (equal to 10 mg/kg of Nar); after 3 h, the mice were challenged with LPS (7.5 mg/kg) to induce ALI pneumonia (Figure 9A). All the mice were euthanized at 24 h post-LPS challenge for analysis. The results showed that the treatment with Nar, especially NPs-loaded Nar, significantly reduced the lung edema and inflammatory cell infiltration compared with the free drug treatment (Figure 9B).

It is well known that preventing or inhibiting the cytokine storm may be one of the key strategies to save the lives of patients with severe pneumonia. Therefore, we examined whether the cytokine storm could be suppressed by targeting CM@Nar-NPs. The levels of IL-6 and IL-1β in plasma were measured by Q-PCR following the treatments (Figure 9C,D). Although free Nar has been demonstrated to decrease the cytokine storm in previous studies [31], the therapeutic efficacy was limited in our experimental mouse ALI model at the dose of 10 mg/kg. The free Nar and Nar-NPs treatment only slightly reduced the cytokine level. Encouragingly, these inflammatory factors were significantly declined after treatment with the CM@Nar-NPs, indicating that the cytokine storm was efficiently calmed down by targeted delivery of Nar using the inflammation-targeting biomimetic nanoparticles. Moreover, a histological assay further validated the existence of excessive pulmonary edema, alveolar inflammatory cell infiltration, and alveolar injury in the PBS-treated septic mice (Figure 9E). In the CM@Nar-NPs-treated group, the inflammatory cell infiltration was reduced more significantly compared with the PBS group and free drug treatment group. Collectively, these results demonstrated that the Nar-NPs, especially the CM@Nar-NPs, effectively inhibited the inflammatory cell infiltration, relieved the pulmonary edema, and calmed down the cytokine storm in the lungs.

### 3.9. Nar Induced Polarization of Macrophages towards M2 Subtype

Macrophages can be affected by various factors to change their phenotype and affect their function. Activated macrophages are polarized into two categories: M1 (F4/80+CD11c+CD206−) macrophages, which are mainly associated with pro-inflammatory responses, and M2 (F4/80+CD11c−CD206+) macrophages, which are mainly associated with anti-inflammatory responses. Thus, modulating the activation state and subtype of macrophages is an effective method for the treatment of diseases. The number of macrophages in septic mice with different treatments was analyzed by flow cytometry at 24 h post-LPS challenge (Figure 10A). We found that all the formulations of the Nar-treated group significantly decreased the number of M1 macrophages, and oppositely increased the number of M2 macrophages in the lungs (Figure 10B,C), suggesting that Nar treatment led to alveolar macrophages polarizing from the M1-type to M2-type. Notably, among all the groups, the CM@Nar-NPs showed the most significant therapeutic effects, which might be attributed to the targeted accumulation of CM@Nar-NPs in the lungs of septic mice. Collectively, these results indicate that CM-coated biomimetic nanoparticles can effectively deliver Nar into the lungs of septic mice and exert anti-inflammatory effects by regulating the level of inflammatory factors to induce alveolar macrophages polarized towards M2.

### 3.10. In Vivo Biosafety of Different Formulations of Nar

To evaluate the in vivo toxicity of different of Nar, the ALI mice were i.t. administrated with Nar for six days (once/day), at 7th day, the mice were euthanasia and the main organs except the lung were extractedand stained with H&E (Figure 11A showed the experimental protocol). The results (Figure 11B) showed that there were no significant differences among all the groups. The results together demonstrated the biosafety of CM@Nar-NPs in the treatment of ALI.

## 4. Discussion

Cytokine storm is characterized by an excessive inflammatory reaction in which proinflammatory cytokines are increasingly released, leading to tissue injury and an unfavorable prognosis in infectious disease. After pathogen invasion, the host immune system is activated, leading to the activation of immune cells such as macrophages. These immune cells can recognize LPS present in Gram-negative bacteria through Toll-like receptors (TLRs). As a result, proinflammatory factors such as TNF-α, IL-1β, and IL-6 are released, leading to damage to the microvascular endothelium, persistent hypotension, and organ failure [32]. Thus, cytokine storm is believed to be one of the major mechanisms which contributes to ALI/ARDS.

Currently, the primary treatment for sepsis relies on the administration of antibiotics. However, the acute physiological changes that occur during sepsis can negatively impact the pharmacokinetics of drugs and hinder their delivery to the disease sites. Thus, developing effective drugs or strategies to prevent or inhibit the cytokine storm may be one of the keys to saving the lives of patients with severe pneumonia [33]. Gu et al. employed a stem cell membrane-cloaked drug-loaded nanoparticle to treat sepsis, showing good therapeutic effects in reducing proinflammatory cytokine levels and protecting against organ damage [19]. Naringin is a promising candidate for clinical use, because it has a wide range of pharmacological activities, including anti-inflammatory, anti-cancer activities, as well as effects on bone regeneration, metabolic syndrome, oxidative stress, genetic damage, and central nervous system (CNS) diseases [34]. In this work, to address the merits of naringin and stem cell biomimetics, we developed a biomimetic Nar-loaded delivery system (CM@Nar-NPs) by combining nanotechnology and cell membrane biomimetic technology. This kind of stem cell membrane-cloaked nanoplatform can efficiently target to lung inflammation and deliver naringin to improve the therapeutic efficacy in the treatment of LPS-induced lung ALI and has showed potential prospects in clinics.

ROS, which is mainly produced by mitochondria, plays vital roles in various cellular functions, including proliferation, differentiation, migration, and apoptosis. Low levels of ROS are very important as redox-signaling molecules in a wide spectrum of pathways are involved in the maintenance of cellular homeostasis and regulating key transcription factors. Additionally, excess ROS could cause organ injury via oxidative damage, inflammatory cascade initiations, and systemic disturbances [35]. ALI-related tissue injury is closely associated with ROS overproduction [36], and the poor prognosis in ALI is attributed to the imbalance between the production and scavenging of ROS [35]. Thus, ROS could be an effective target to fight against inflammatory diseases. In recent years, some antioxidant agents have been developed to treat ROS-related diseases including ALI [37]. Although small molecular compounds present excellent anti-inflammation effects by eliminating local ROS, the properties of these kinds of compounds, such as off-targeting and being easily cleared out by system, determined that it must require high doses to obtain effective therapeutics. Nar is a naturally occurring flavonoid commonly found in grapefruits and other kinds of the Citrus genus, which has been used in traditional Chinese medical regimens for thousands of years. Nar exhibits excellent pharmacological effects, such as anti-inflammation, antioxidant, lipid-lowering, etc. [34]. However, the poor solubility and lower biocompatibility of Nar greatly limit its therapeutic efficiency. In Li’s review paper [38], they concluded that the doses of free Nar to treat acute lung injury in mice in previous studies were about 10, 20, or 40 mg/kg body weight. Due to the poor water solubility of Nar, the high dose of Nar means more DMSO in the solution; the *i.t*. administration might damage the respiratory mucosa. Thus, 10 mg/kg was determined in this study. Additionally, our cell viability assay showed that 10 µg/mL of Nar or Nar-NPs would not inhibit the cell growth of the RAW264.7 cells and HUVECs, while, exceeding 10 µg/mL of all the formulations of Nar could inhibit the cell growth at varying degrees. Based on the cell result, we also confirmed the dose of 10 mg/kg in animal studies.

Here, we employed stem cell membrane-cloaked PLGA nanoparticles to deliver Nar to the sites of inflammatory lungs. This kind of nano-platform is expected to possess both the inflammatory targeting of stem cells and the small-size effect of nanoparticles [39]. The characterization results of our biomimetic nanoparticle showed that with the coating of stem cell membranes, the CM@Nar-NPs showed a good dispersion compared with the easier-aggregating property of the Nar-NPs (Figure 2). Also, both the Nar-NPs and CM@Nar-NPs exhibited improved biocompatibility and biosafety in vivo. As expected, the CM@Nar-NPs could effectively evade the phagocytosis and clearance by the immune system and successfully target and accumulate in the lungs of septic mice (Figure 1).

The vascular endothelium plays an important role in the migration of macromolecules and inflammatory cells from the blood to tissue. Under the inflammatory conditions, the overproduction of ROS leads to the opening of inter-endothelial junctions and promotes the migration of inflammatory cells across the endothelial barrier. The migrated inflammatory cells not only promote the clearance of pathogens and foreign particles but also result in tissue injury. Persistently increased lung microvascular permeability resulting in protein-rich lung edema is a hallmark of ALI pneumonia [40,41]. In this study, we observed that the Nar-NPs, especially the CM@Nar-NPs intranasal treatment, significantly inhibited lung vascular injury and inflammation following LPS challenge. The data indicated that the CM@Nar-NPs treatment resulted in a marked decrease in lung vascular permeability as determined by EBA extravasation assay (Figure 6). Also, the mice treated with the CM@Nar-NPs exhibited a marked increase in survival rate (Figure 7). Thus, targeting microvascular leakiness to restore lung fluid homeostasis is a potential therapeutic approach for the prevention and treatment of pneumonia.

Inflammation could lead to overproduction of ROS, which causes mitochondrial dysfunction, irreversible cell damage, and even cell death [26]. Therefore, effectively regulating the balance of intracellular ROS is of great importance in controlling disease progression. Here, the RAW264.7 cells were pretreated with 80 μg/mL of different formulations of Nar for 1 h, and then treated with 1 μg/mL of LPS (the main pathogenic components of Gram-negative bacteria) for 6 h to induce inflammation and ROS overproduction. With LPS stimulation, the cells produced an excessive amount of ROS, as evidenced by the high MFI of DCFH-DA. Interestingly, the MFI was markedly decreased after incubation with all formulations of Nar, among which, the effect of the CM@Nar-NPs represented the most significant curative effect. Simultaneously, the ROS level of the lungs and livers of septic mice were determined using DCFH-DA staining and IVIS imaging system. The in vivo results were consistent with the ROS observed in the RAW264.7 cells (Figure 6). Collectively, the Nar-NPs, especially the CM@Nar-NPs, exhibited the capability to scavenge ROS in both inflammatory cells and mice.

Based on previous reports [2,42], inhibiting or calming down the cytokine storm is one of the key methods to save the lives of patients with severe pneumonia. In this study, although Nar has been reported to decrease the cytokine storm [43] at 200 mg/kg, the therapeutic efficacy was limited in our experimental mouse ALI model at the dose of 10 mg/kg. An amount of 10 mg/kg of free Nar and Nar-NPs treatment moderately reduced the immune cell infiltration and the levels of IL-6 and IL-1β. Encouragingly, these inflammatory factors were significantly decreased following the treatment with the CM@Nar-NPs at the same dose of 10 mg/kg (Figure 9). This suggested that the anti-inflammation efficacy was significantly improved with the targeted delivery by the stem cell membrane-cloaked biomimetic nanoplatform.

Macrophages can be adjusted by a variety of factors to change their phenotype and function. Activated macrophages are mainly polarized into two categories, M1-like macrophages and M2-like macrophages. M1 macrophages are pro-inflammatory and have a central role in host defense against infection, while M2 macrophages are associated with anti-inflammatory and tissue remodeling responses. Re-polarization of M1-type macrophages into M2-type macrophages is a promising treatment for inflammation [39]. Our data demonstrated that the infiltrated macrophages enhanced M1 phenotype polarization in LPS-treated septic mice, and enhanced M2 phenotype polarization after treatment with Nar, indicating a shift in the macrophage polarization towards the M2 subtype (Figure 10). This result suggested that effects of Nar or CM@Nar-NPs in treating inflammation diseases might be related to M2 polarization with anti-inflammatory activity.

Although cell membrane biomimetics technology has not yet achieved comprehensive clinical implementation, it is believed that the cell membrane biomimetics technology will open new doors in the field of nanomedicine. However, the excessive use of immune cell membrane-coated nanoparticles may induce or exacerbate inflammation through interactions with the immune system, potentially leading to the release of pathological mediators. Thus, there is still a lot of work to be explored before their clinical application.

## 5. Conclusions

In summary, we designed a stem cell membrane-cloaked nanoplatform to deliver anti-inflammation drugs for pneumonia treatment. The as-synthesized CM@Nar-NPs exhibited excellent dispersion, good biocompatibility, and increased biosafety in vitro and in vivo. This biomimetic drug delivery platform could evade the clearing by the immune system and successfully accumulate into inflammatory lungs in septic mice. In the LPS-induced ALI mouse model, the CM@Nar-NPs significantly improved the survival rate, calmed down the cytokine storm in BALF and lungs, suppressed the vascular permeability of lung edema, and decreased the inflammatory cell infiltration. Furthermore, the flow cytometry analysis suggested that the enhanced therapeutic effect of the CM@Nar-NPs was through promoting macrophage polarization from the M1 to M2 subtype. Notably, by loading into the nanoplatform, the effective dose of Nar in this study (10 mg/kg) decreased more than 10-fold in comparison with the effective dose reported in previous studies (100–200 mg/kg). Our study highlights that targeted drug delivery to treat pneumonia significantly reduces the cytokine storm syndromes compared with free drug therapy. Our strategy may provide a promising nanoplatform for targeting treatment of pneumonia, and this strategy shows potential clinical application prospects.

## Figures and Tables

**Figure 1 antioxidants-13-00282-f001:**
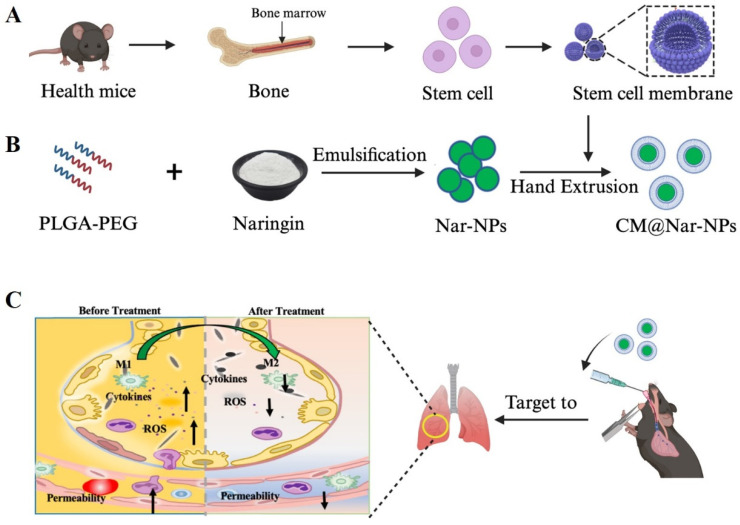
Schematic illustration of the preparation of Nar-loaded PLGA nanoparticles cloaked with stem cell membrane (CM@Nar-NPs) and their treatment for ALI. (**A**) Cell membrane vesicles (CM) were extracted from the mesenchymal stem cells isolated from healthy mouse bone marrow. (**B**) Nar-loaded PLGA NPs were prepared using emulsification–evaporation method. Then, the harvested Nar-NPs were mixed with the CM and extruded through a polycarbonate porous membrane to form CM@Nar-NPs. (**C**) The proposed illustration of tracheal instillation (*i.t.*) administered ((CM@Nar-NPs for the targeting and improving of ALI treatment through inhibition of ROS-production and cytokine-secrection in the inflammed lungs. And, Nar-induced macrophages towards the M2 subtype which might contribute to its anti-inflammation effects. (The up arrows show the increase and down arrows show the decrease tendency).

**Figure 2 antioxidants-13-00282-f002:**
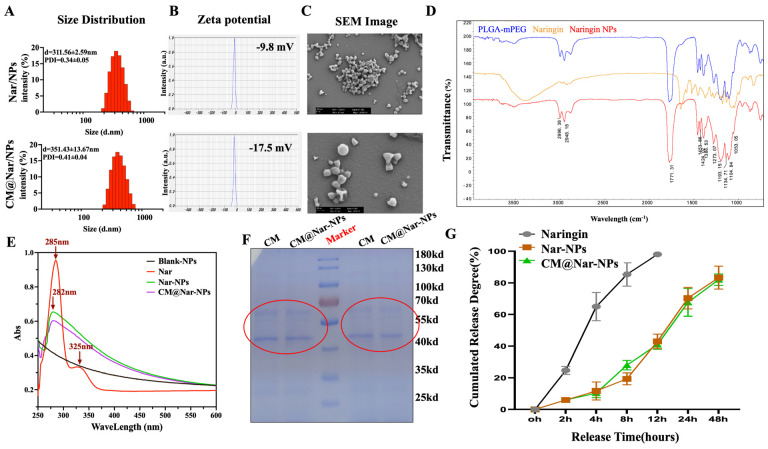
**Characterizations of CM@Nar-NPs.** (**A**,**B**) Size distribution, polydispersity index (PDI), and zeta potential of Nar-NPs and CM@ Nar-NPs, respectively. (**C**) Representative morphology image of Nar-NPs and CM@ Nar-NPs determined by scanning electron microscope (1 bar = 200 nm). (**D**) FTIR analysis of PLGA-PEG, Nar, and Nar-NPs. (**E**) UV–vis spectrum of free Nar and CM@Nar-NPs indicated that Nar was encapsulated in CM@Nar-NPs, which was shown by the similar absorption peaks at around 210 nm. (**F**) SDS-PAGE protein analysis of stem cell membrane (CM) and CM@Nar-NPs (the samples were determined at the same protein concentrations), red circles show the biomimetic NPs retained the characteristic proteins of CM. (**G**) In vitro release of naringin from NPs in PBS (0.01 M, PH = 7.4, *n* = 3/group).

**Figure 3 antioxidants-13-00282-f003:**
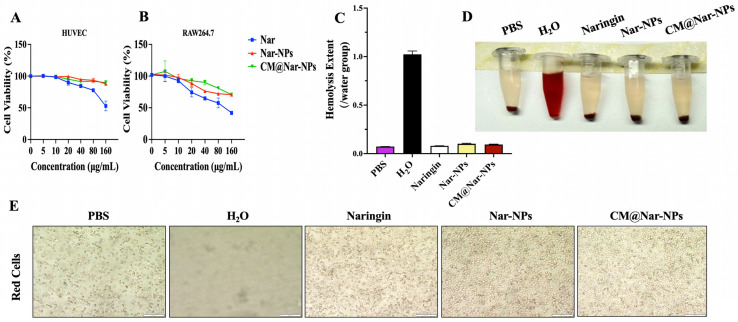
**In vitro biosafety of different formulations of Nar.** (**A**,**B**) Cell viability of RAW264.7 and HUVECs treated with different concentration of Nar for 24 h. (**C**,**D**) The images and quantification analysis of hemolysis assay of Nar, Nar-NPs, and CM@Nar-NPs. (**E**) Representative micrographs of main organ cross-sections of healthy mice *i.n.* administered with different formulations of Nar for one week (once/day), which was stained by H&E. 1 bar = 200 µm.

**Figure 4 antioxidants-13-00282-f004:**
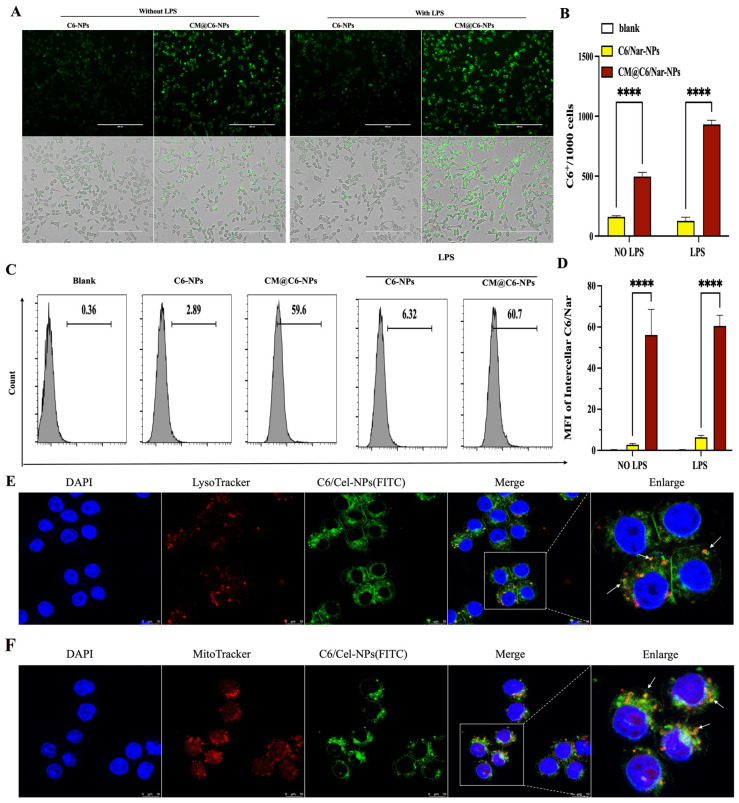
**In vitro targeting and biodistribution of CM@Nar-NPs in RAW264.7 cells.** Due to the weak fluorescence of Nar, Coumarin 6 (C6) was co-loaded into the NPs as fluorescent marker. (**A**,**B**) Cellular uptake of free C6/Nar or CM@C6/Nar-NPs in RAW264.7 cells co-cultured with NPs for 2 h: Fluorescence images (**A**) and quantitative analysis (**B**) of cellular uptake in RAW264.7 cells without or with LPS stimulation. Scale bars: 200 μm. (**C**,**D**) The typical images (**C**) and quantitative analysis of MFI of intracellular C6 (**D**) determined by flow cytometry. **** *p* < 0.0001 versus LPS group (one-way ANOVA with Bonferroni post hoc test). (**E**,**F**) Biodistribution of CM@C6/Nar-NPs in LPS-induced RAW264.7 cells: Lysosomes co-localization (**E**) of CM@C6/Nar-NPs (**E**) and mitochondria co-localization(**F**) of CM@C6/Nar-NPs. Scale bars: 10 μm. (White arrows indicate the co-localization of nanparticles and lysosomes or mitochondria).

**Figure 5 antioxidants-13-00282-f005:**
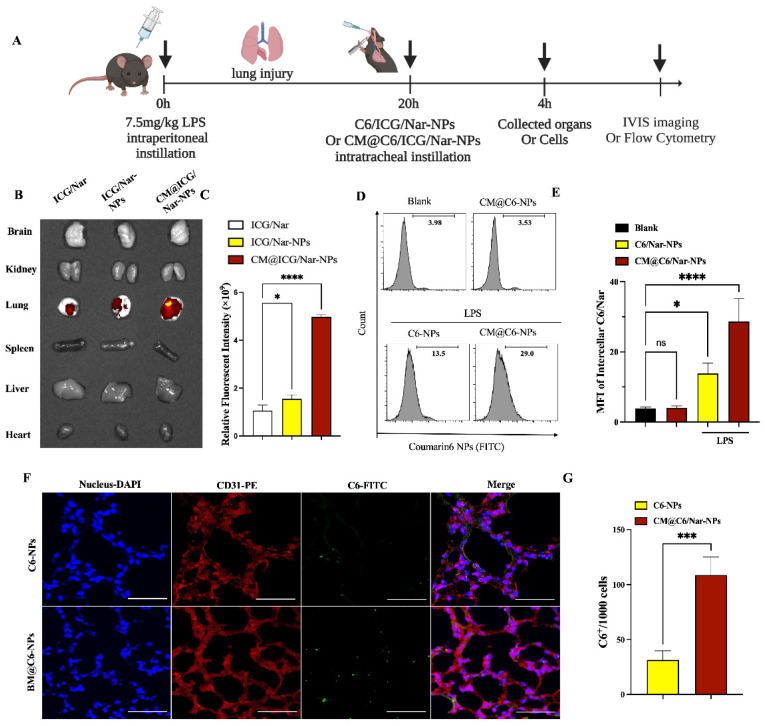
**In vivo targeting and cellular uptake of different formulations of Nar in lungs of ALI mice.** Indocyanine green (ICG) was co-loaded into the NPs as a fluorescent marker in vivo. (**A**) Time axis of in vivo imaging determination: the mice were *i.p.* administered with LPS (7.5 mg/kg body weight) to induce a sepsis model, the mice were *i.t.* administered with different formulations of Nar at 20 h post LPS challenge. The mice were sacrificed and the main organs collected at 24 h post LPS challenge. (**B**) Typical fluorescence imaging of the main organs originated from ALI mice after *i.t.* administration of CM@Nar/ICG-NPs for 4 h. (**C**) Relative accumulations of Nar/ICG-NPs in lung tissues measured with Living Image 4.5 software, which demonstrated that CM cloaked NPs significantly promoted the accumulation of Nar/ICG in lungs of ALI mice. (**D**) In vivo cellular uptake of NPs in lung cells assayed by flow cytometry. (**E**) Semi-quantitative analysis of MFI (mean fluorescence intensity) of intracellular C6. (**F**,**G**) The fluorescent pictures and quantitative analysis of C6/Nar-NPs or CM@C6/Nar-NPs (green color) in lung sections at 4 h after *i.t.* administration with free C6/Nar or CM@C6/Nar-NPs (blue color of DAPI to locate the nucleus). Data are expressed as mean ± SD (*n* = 4). * *p* < 0.05, *** *p* < 0.001, **** *p* < 0.0001 versus ICG/Nar group and ns means no significance (one-way ANOVA with Bonferroni post hoc test).

**Figure 6 antioxidants-13-00282-f006:**
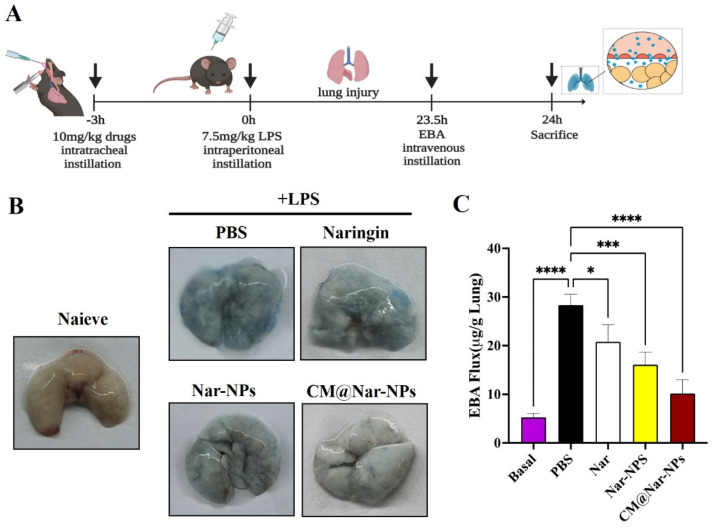
**Inhibition of lung vascular permeability in ALI mice treated with different formulations of Nar.** (**A**) Schematic illustration of vascular permeability assayed by Evans Blue: firstly, the mice were *i.t.* administered with different formulations of Nar (10 mg/kg); after 3 h, the mice were *i.p.* administered with LPS (7.5 mg/kg body weight). Evans blue album (EBA) was *i.v.* injected at 23.5 h post LPS challenge. The mice were sacrificed and the lungs were collected at 24 h post LPS challenge to test the EBA level in the lungs. (**B**) Representative images of lung tissues showed that Nar-NPs, especially CM@Nar-NPs could significantly decrease the lung vascular injury. Nar, Nar-NPs, and CM@Nar-NPs were *i.t.* administered (10 mg/kg, *i.n.*, single dose) to mice at 3 h pre-LPS challenge. At 24 h post-LPS challenge, the mice were euthanized and lungs were perfused to remove blood and imaged. (**C**) The amount of Evans blue (EB) fluxed in lungs was measured at 620 nm to evaluate the permeability of lung vessels. Data are expressed as mean ± SD (*n* = 4). * *p* < 0.05, *** *p* < 0.001 and **** *p* < 0.0001 versus PBS-treated group (one-way ANOVA with Bonferroni post hoc test).

**Figure 7 antioxidants-13-00282-f007:**
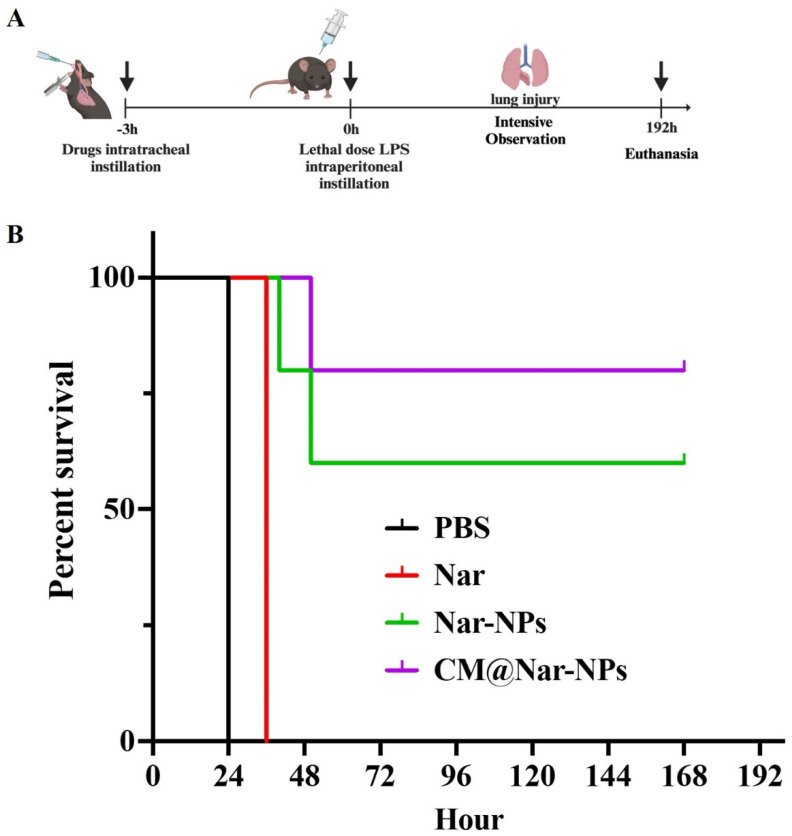
**Survival rates of mice.** (**A**) Time axis of survival rate determination: the mice were *i.t.* administered with different formulations of Nar for a single dose; after 3 h, the mice were *i.p.* administered with a lethal dose of LPS (10 mg/kg body weight). Then, the mice were kept under intense observation and the survival was recorded. (**B**) Mice were pretreated with different formulations of Nar (10 mg/kg) 3 h before intraperitoneal injection with LPS at a lethal dose (12 mg/kg body weight). All the mice were under close observation for 7 days to assay the survival rate (*n* = 6/group).

**Figure 8 antioxidants-13-00282-f008:**
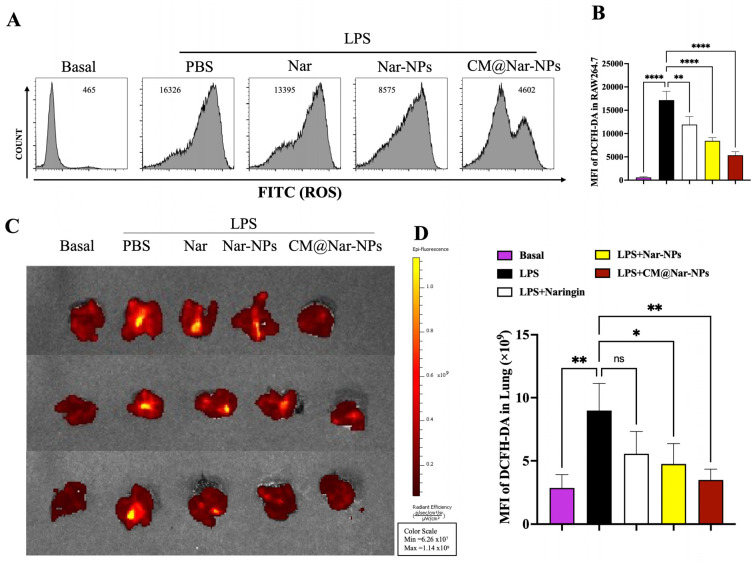
**ROS scavenging activity of Nar determined by DCF-HA staining in vitro and in vivo.** Raw264.7 cells and lungs with different treatments were stained with DCF-HA kits and determined by flow cytometry or the IVIS imaging system, respectively. (**A**,**B**) Flow cytometry (FCM) analysis of ROS level in Raw264.7 cells challenged with LPS (500 ng/mL) for 24 h. Raw264.7 cells pretreated with different formulations of Nar (10 mg/kg) 3 h before 1 μg/mL of LPS challenging; the mean fluorescent intensity (MFI) of 20,000 cells was measure using FCM at the FITC channel. (**C**) The fluorescent pictures of lungs originated from differently treated mice stained with DCF-HA kits imaged by an IVIS imaging system. (**D**) Relative MFI of DCF-HA in lung tissues of (**C**) measured with Living Image 4.5 software. Data are expressed as mean ± SD (*n* = 4). * *p* < 0.05, ** *p* < 0.01 and **** *p* < 0.0001 versus PBS-treated group, ns means no significance (one-way ANOVA with Bonferroni post hoc test).

**Figure 9 antioxidants-13-00282-f009:**
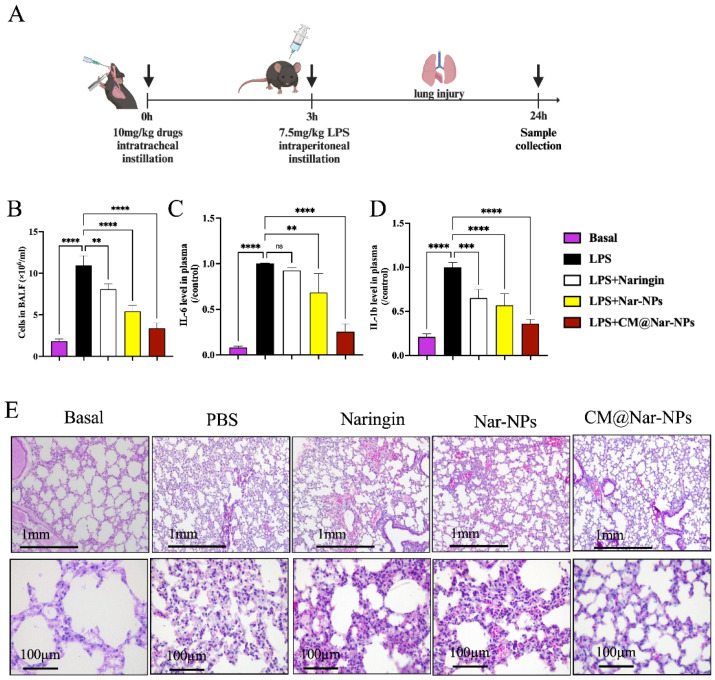
**The anti-inflammation capacity of different formulations of Nar.** (**A**) Time axis of therapeutics determination: The mice were pretreated with different formulations of Nar (10 mg/kg) 3 h before intraperitoneal injection with LPS (7.5 mg/kg). Bronchoalveolar fluid (BALF), blood, and lung tissues were collected at 24 h post-LPS challenge. (**B**) Total cell counts in BALF. (**C**,**D**) Expression levels of IL-6 and IL-1β in mouse plasma at 24 h post-LPS challenge. Data are expressed as mean ± SD (*n* = 4). ** *p* < 0.01, *** *p* < 0.001 and **** *p* < 0.0001 versus PBS-treated group, ns means no significance (one-way ANOVA with Bonferroni post hoc test). (**E**) Representative micrographs of lung tissue cross-sections at 24 h post-LPS challenge stained by H&E. Scale bar: 1 mm (**upper row**) or 100 μm (**lower row**).

**Figure 10 antioxidants-13-00282-f010:**
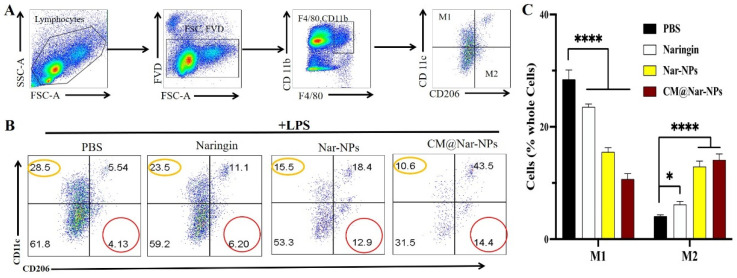
**CM@Nar-NP reduced lung inflammation by inhibiting the activation of M1 macrophages and inducing M2 polarization.** Flow cytometric analysis of macrophages derived from inflammatory lungs treated with different formulations of Nar for 24 h. (**A**) The selection of M1- or M2-positive cells. In brief, we first gated the live cells, and then gated the CD11b+F4/80+ cells. In this subgroup, we used CD11c and CD206 as markers to identify M1 or M2 macrophages, respectively. CD11b+F4/80+, CD11c+, CD206− cells are zoned as M1 cells, CD11b+F4/80+, CD11c−, CD206+ cells are marked as M2 cells. (**B**) The representative images of cell distribution pattern in sepsis mice treated with different formulations of Nar. (The percentages of M1 and M2 cells were shown in yellow circles and red circles, respectively. (**C**) The quantitative analysis of percentage of total M1 macrophages and M2 macrophages in sepsis lungs. Data are expressed as mean ± SD (*n* = 4). * *p* < 0.05 and **** *p* < 0.0001 versus PBS-treated group (one-way ANOVA with Bonferroni post hoc test).

**Figure 11 antioxidants-13-00282-f011:**
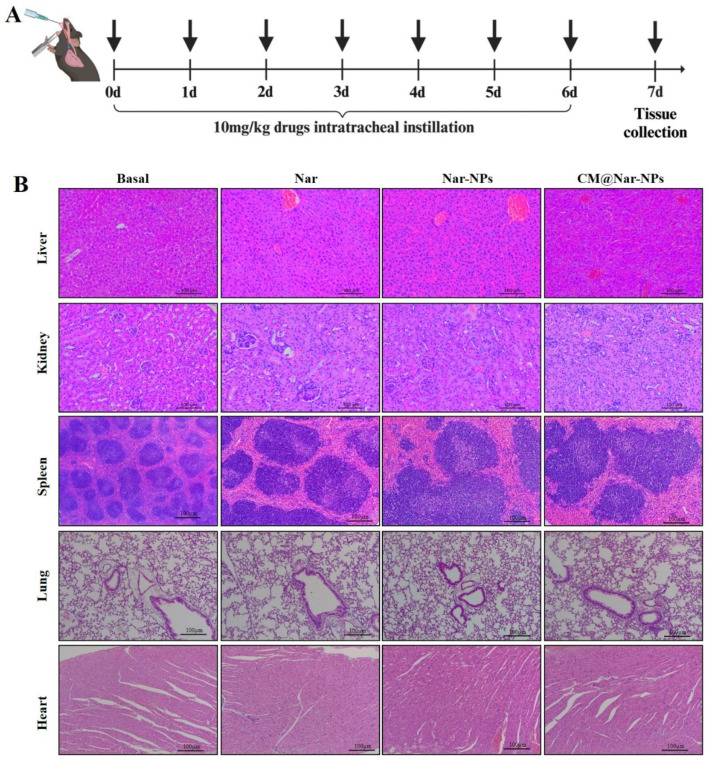
**In vivo biosafety validation of different formulations of Nar.** (**A**) Time axis of biosafety determination: The mice were *i.t.* administered with different formulations of Nar (10 mg/kg) once a day for 7 days. On 8th day, the mice were euthanized and the main organs were harvested for tests. (**B**) H&E staining images of the major organs (lung, liver, spleen, kidney, and heart) collected from the mice after *i.t.* administration of Nar, which showed no obvious damage or abnormalities in the main organs in all the three formulations of Nar.

## Data Availability

The datasets used and/or analyzed during the current study are available from the corresponding author on reasonable request.

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
