# Peer review of "Intratracheal Administration of Stem Cell Membrane-Cloaked Naringin-Loaded Biomimetic Nanoparticles Promotes Resolution of Acute Lung Injury"

_antioxidants, 2024, doi:10.3390/antiox13030282_

Round 1
Reviewer 1 Report
This work from Hua Jin et al investigates the role of naringin nanoparticles on the resolution of ALI. There is much merit to this study but a few issues still need to be addressed for the improvement of the manuscript.
Please, in the abstract section, write the full name for “CRS”.
To help the reader to understand the aim of the study, please, add the aim in the Introduction.
Figure 2 and 5, the letter font is too small.
Figure 3, graph image is in a poor quality and the letter font is also too small to read.
Results section, line 392, indicate the figure that should represent the data.
The authors discussed that ALI induces neutrophil infiltration, to support better the conclusion and title, would be interesting to have IL-8 protein levels for the in vivo study.
Author Response
Detail comments
Please, in the abstract section, write the full name for “CRS”.
Response: Thanks for your suggestion. We have added the full name for “CRS” as “cytokine storm release syndrome” in the abstract part.
To help the reader to understand the aim of the study, please, add the aim in the Introduction.
Response: Thanks for your suggestion. We have added the aim as shown in the green words in the introduction part.
Figure 2 and 5, the letter font is too small.
Figure 3, graph image is in a poor quality and the letter font is also too small to read.
Response: Thanks for these comments. We have fixed these problems and provided the images with high resolution in the revised figures.
Results section, line 392, indicate the figure that should represent the data.
Response: Per your suggestion, we have revised the description in the revised manuscript.
The authors discussed that ALI induces neutrophil infiltration, to support better the conclusion and title, would be interesting to have IL-8 protein levels for the in vivo study.
Response: Thanks for this comment. In the current study, we mainly focus on the macrophage polarization and the changes of IL-6 level after treated with Nar. Your suggestion is very helpful for investigating the neutrophil and the potential IL-8 expression, which we will look carefully in our future study.

Reviewer 2 Report
Comments and Suggestions for Authors
The study focuses on using naringin-loaded nanoparticles, coated with mesenchymal stem cell membranes, for targeted delivery to inflamed lungs. The paper is well-structured, presenting a clear abstract, introduction, materials and methods, results, and a discussion section. The research is significant as it addresses a critical medical condition with a novel treatment strategy.
However, there are several areas that require attention before the manuscript can be considered for publication:
-
- Certain sections of the methodology require more detail to ensure reproducibility and clarity in understanding the procedures followed.
-
- Paper would benefit from a more detailed explanation of the statistical methods used, including any assumptions and the rationale for choosing specific tests.
-
-The discussion section could be enhanced by acknowledging the limitations of the study, which helps in contextualizing the results and potential areas for future research.
-
-The paper would be strengthened by a more detailed comparative analysis with existing treatments or approaches for ALI/ARDS, which would highlight the novelty and potential impact of the study's findings.
-
- I found several english errors throughout the manuscript. Please have a deep language revision.
-
Moderate revision required
Author Response
- - Certain sections of the methodology require more detail to ensure reproducibility and clarity in understanding the procedures followed.
Thanks for your suggestions. We have provided more details in the methods part (shown by the blue words).
- - Paper would benefit from a more detailed explanation of the statistical methods used, including any assumptions and the rationale for choosing specific tests.
Response: Thanks for your helpful comment. We have added all the statistical details in the revised Figure legends.
- -The discussion section could be enhanced by acknowledging the limitations of the study, which helps in contextualizing the results and potential areas for future research.
Response: Thanks for this comment. We have discussed this issue in the revised discussion part (shown in blue words).
- -The paper would be strengthened by a more detailed comparative analysis with existing treatments or approaches for ALI/ARDS, which would highlight the novelty and potential impact of the study's findings.
Response: Thanks for this helpful comment. We have added certain background description in the Instruction part (shown in blue words, in Page 3).
- - I found several english errors throughout the manuscript. Please have a deep language revision.
Response: Thanks for your suggestion. We have gone through the full-text and revised the errors (shown the colored words in manuscript).

Reviewer 3 Report
The article covers a very interesting and current topic. Nevertheless, in my opinion, some parts need to be improved, I have some comments:
1) Abstract. In ALI mouse model, intratracheal instillation (i.t) of CM@Nar-NPs significantly decreased the ROS level, inhibited the proinflammatory cytokines, and remarkably promoted survival rate. Additionally, CM@Nar-NPs increased the expression of M2 marker (CD206), and decreased the expression of M1 marker (F4/80) in septic mice, suggesting that Nar-modulated macrophage polarized towards M2 subtype might contribute to their anti-inflammation effects. This work proves that mesenchymal stem cell membrane-based biomimetic nanoparticle delivery system could efficiently target lung inflammation via i.t administration. This nano-system provides an excellent i.t platform for targeted pneumonia treatment. Please, improve the results to support the conclusions.
2) Abstract. In ALI mouse model, intratracheal instillation (i.t) of CM@Nar-NPs significantly decreased the ROS level, inhibited the proinflammatory cytokines, and remarkably promoted survival rate. Additionally, CM@Nar-NPs increased the expression of M2 marker (CD206), and decreased the expression of M1 marker (F4/80) in septic mice, suggesting that Nar-modulated macrophage polarized towards M2 subtype might contribute to their anti-inflammation effects. This work proves that mesenchymal stem cell membrane-based biomimetic nanoparticle delivery system could efficiently target lung inflammation via i.t administration. This nano-system provides an excellent i.t platform for targeted pneumonia treatment. Abstract might be beneficial to include a sentence that briefly summarizes the key findings of the study. This can provide readers with a quick overview of the research.
3) 1. Introduction L34-39. Cytokine storm is an overactivation of the immune system caused by infections, drugs, etc., which can occur rapidly to lead to organ failure, life-threatening, even death [1]. In the Covid-19 epidemic, most patients with COVID-19 developed acute respiratory distress syndrome (ARDS) [2], multiple organ failure [3] and even death. It is noticeable that one of the most important mechanisms underlying the deterioration of this disease is cytokine storm release syndrome (CRS). Although the Authors described in detail the findings from the included references, there are several relevant works/reviews, including most recently published which should be added and discussed by the Authors:
a-Short Survey on the Protein Modifications in Plasma during SARS-CoV-2 Infection. Int J Mol Sci. 2023 Sep 14;24(18):14109. doi: 10.3390/ijms241814109.
b- The Efficacy of Antioxidant Oral Supplements on the Progression of COVID-19 in Non-Critically Ill Patients: A Randomized Controlled Trial. Antioxidants 2021, 10, 804. https://doi.org/10.3390/antiox10050804
4) In this work, the naringin nanoparticles (Nar-NPs) were prepared by emulsification-and-evaporation method, and then the mesenchymal stem cell membranes (CM) were extracted and coated onto the surface of Nar-NPs through hand extrusion method to obtain the biomimetic CM@Nar-NPs. In vitro, CM@Nar-NPs represent good dispersity, excellent biocompatibility and biosafety. At the cellular level, CM@Nar-NPs showed excellent abilities of targeting to the inflamed macrophages and capacity of scavenging ROS. In vivo imaging demonstrated that CM@Nar-NPs could target and accumulate into the inflammatory lungs. In ALI mouse model, intratracheal instillation (i.t) of CM@Nar-NPs significantly decreased the ROS level, inhibited the proinflammatory cytokines, and remarkably promoted survival rate. Additionally, CM@Nar-NPs increased the expression of M2 marker (CD206), and decreased the expression of M1 marker (F4/80) in septic mice, suggesting that Nar-modulated macrophage polarized towards M2 subtype might contribute to their anti-inflammation effects. This work proves that mesenchymal stem cell membrane-based biomimetic nanoparticle delivery system could efficiently target lung inflammation via i.t administration. This nano-system provides an excellent i.t platform for targeted pneumonia treatment. Please improve the description of this part and underline the novelty of the study.
5) 2.20. Statistical Analysis 279 Results are expressed as mean ± SEM Statistical significance was determined by one- 280 way ANOVA with a Games-Howell post hoc analysis for multiple-group comparisons. 281 Two-group comparisons were analyzed by the two-tailed unpaired Student’s t-test. Please, improve the description of statistical tests used to evaluate the data.
6) 4. Discussion L608-611. Cytokine storm is characterized by excessive inflammatory reaction in which proin- 609 flammatory cytokines are increasingly released, leading to tissue injury and an unfavorable prognosis in infectious disease. Cytokine storm was believed to be one of the major mechanisms which contribute to ALI/ARDS. The discussion section needs to be improved. It is necessary to be more concise in the presentation of the facts, clarifying the results obtained and comparing them with previous or similar studies.
7) Figure 11. In vivo biosafety validation of different formulations of Nar. (A) Time-axis of biosafety 603 determination: The mice were i.t administrated with different formulations of Nar (10mg/kg) once 604 a day for 7 days. Bronchoalveolar Fluid (BALF), blood, and lung tissues were collected at 7th day. 605 (B) H&E staining images of the major organs (lung, liver, spleen, kidney and heart) collected from 606 the mice after i.t. administration of Nar. Please, improve the description of the figure.
No further comments. All detail comments are reported above.
Author Response
- In ALI mouse model, intratracheal instillation (i.t) of CM@Nar-NPs significantly decreased the ROS level, inhibited the proinflammatory cytokines, and remarkably promoted survival rate. Additionally, CM@Nar-NPs increased the expression of M2 marker (CD206), and decreased the expression of M1 marker (F4/80) in septic mice, suggesting that Nar-modulated macrophage polarized towards M2 subtype might contribute to their anti-inflammation effects. This work proves that mesenchymal stem cell membrane-based biomimetic nanoparticle delivery system could efficiently target lung inflammation via i.t administration. This nano-system provides an excellent i.t platform for targeted pneumonia treatment. Please, improve the results to support the conclusions.
Response: Thanks a lot for your suggestions! We have improved the resolution of certain figures and the description of the results as shown in colored words in this revised manuscript.
- In ALI mouse model, intratracheal instillation (i.t) of CM@Nar-NPs significantly decreased the ROS level, inhibited the proinflammatory cytokines, and remarkably promoted survival rate. Additionally, CM@Nar-NPs increased the expression of M2 marker (CD206), and decreased the expression of M1 marker (F4/80) in septic mice, suggesting that Nar-modulated macrophage polarized towards M2 subtype might contribute to their anti-inflammation effects. This work proves that mesenchymal stem cell membrane-based biomimetic nanoparticle delivery system could efficiently target lung inflammation via i.t administration. This nano-system provides an excellent i.t platform for targeted pneumonia treatment. Abstract might be beneficial to include a sentence that briefly summarizes the key findings of the study. This can provide readers with a quick overview of the research.
Response: Thanks a lot for your helpful suggestion. We have revised the abstract (show in red words in the abstract part) as following:
Collectively, this work proves that mesenchymal stem cell membrane-based biomimetic nanoparticle delivery system could efficiently target lung inflammation via i.t administration, and then the released payload inhibited the production of inflammatory cytokines and ROS, and Nar-modulated macrophage polarized towards M2 phenotype might contribute to their anti-inflammation effects. This nano-system provides an excellent pneumonia treated platform with satisfied biosafety and have great potential to effectively deliver herbal medicine.
3) 1. Introduction L34-39. Cytokine storm is an overactivation of the immune system caused by infections, drugs, etc., which can occur rapidly to lead to organ failure, life-threatening, even death [1]. In the Covid-19 epidemic, most patients with COVID-19 developed acute respiratory distress syndrome (ARDS) [2], multiple organ failure [3] and even death. It is noticeable that one of the most important mechanisms underlying the deterioration of this disease is cytokine storm release syndrome (CRS). Although the Authors described in detail the findings from the included references, there are several relevant works/reviews, including most recently published which should be added and discussed by the Authors:
a-Short Survey on the Protein Modifications in Plasma during SARS-CoV-2 Infection. Int J Mol Sci. 2023 Sep 14;24(18):14109. doi: 10.3390/ijms241814109.
b- The Efficacy of Antioxidant Oral Supplements on the Progression of COVID-19 in Non-Critically Ill Patients: A Randomized Controlled Trial. Antioxidants 2021, 10, 804. https://doi.org/10.3390/antiox10050804
Response: Thanks for your suggestion. We have added the references in the revised manuscript.
4) In this work, the naringin nanoparticles (Nar-NPs) were prepared by emulsification-and-evaporation method, and then the mesenchymal stem cell membranes (CM) were extracted and coated onto the surface of Nar-NPs through hand extrusion method to obtain the biomimetic CM@Nar-NPs. In vitro, CM@Nar-NPs represent good dispersity, excellent biocompatibility and biosafety. At the cellular level, CM@Nar-NPs showed excellent abilities of targeting to the inflamed macrophages and capacity of scavenging ROS. In vivo imaging demonstrated that CM@Nar-NPs could target and accumulate into the inflammatory lungs. In ALI mouse model, intratracheal instillation (i.t) of CM@Nar-NPs significantly decreased the ROS level, inhibited the proinflammatory cytokines, and remarkably promoted survival rate. Additionally, CM@Nar-NPs increased the expression of M2 marker (CD206), and decreased the expression of M1 marker (F4/80) in septic mice, suggesting that Nar-modulated macrophage polarized towards M2 subtype might contribute to their anti-inflammation effects. This work proves that mesenchymal stem cell membrane-based biomimetic nanoparticle delivery system could efficiently target lung inflammation via i.t administration. This nano-system provides an excellent i.t platform for targeted pneumonia treatment. Please improve the description of this part and underline the novelty of the study.
Response: Thanks for this comment. Per your suggestion, we have revised and polished the abstract (shown in red words in Abstract) to enhance the novelty of our study.
5) 2.20. Statistical Analysis 279 Results are expressed as mean ± SEM Statistical significance was determined by one-way ANOVA with a Games-Howell post hoc analysis for multiple-group comparisons. 281 Two-group comparisons were analyzed by the two-tailed unpaired Student’s t-test. Please, improve the description of statistical tests used to evaluate the data.
Response: Thanks for this comment, and we have provided all the statistical details in the revised Figure legends.
6) 4. Discussion L608-611. Cytokine storm is characterized by excessive inflammatory reaction in which proin-flammatory cytokines are increasingly released, leading to tissue injury and an unfavorable prognosis in infectious disease. Cytokine storm was believed to be one of the major mechanisms which contribute to ALI/ARDS. The discussion section needs to be improved. It is necessary to be more concise in the presentation of the facts, clarifying the results obtained and comparing them with previous or similar studies.
Response: Thanks for these comments. We have modified certain sentences in the manuscript shown in colored words and compared our study with previous studies by providing more detail discussions in the revised discussion part (shown in red words) as following: After pathogen invasion, the host immune system is activated, leading to the activation of immune cells such as macrophages. These immune cells can recognize lipopolysaccharides present on gram-negative bacteria through Toll-like receptors (TLRs). As a result, proinflammatory factors such as TNF-α, IL-1β, and IL-6 are released, leading to damage to the microvascular endothelium, persistent hypotension, and organ failure [Nat Rev Immunol. 2017;17:407–20.]. Thus, cytokine storm was believed to be one of the major mechanisms which contribute to ALI/ARDS.
Currently, the primary treatment for sepsis relies on the administration of antibiotics. However, the acute physiological changes that occur during sepsis can negatively impact the pharmacokinetics of drugs and hinder their delivery to the disease sites. Thus, to develop effective drugs or strategies to prevent or inhibit the cytokine storm may be one of the keys to saving the life of patients with severe pneumonia [29]. Gu, et al. employed a stem cell membrane-cloaked drug-loaded nanoparticle to treat sepsis, showing good therapeutic effects in reducing proinflammatory cytokine levels and protecting organ damage [doi: 10.1186/s12951-023-01913-3.]. Naringin is a promising candidate for clinical use, because it has a wide range of pharmacological activities, including anti-inflammatory, anti-cancer activities, as well as effects on bone regeneration, metabolic syndrome, oxidative stress, genetic damage and central nervous system (CNS) diseases [doi: 10.1080/13880209.2016.1216131.]. In this work, to address the merits of naringin and stem cell biomimetics, we developed a biomimetic Nar-loaded delivery system (CM@Nar-NPs) by combining nanotechnology and cell membrane biomimetic technology. This kind of stem cell membrane-cloaked nanoplatform can efficiently targeting to lung inflammation and deliver Naringin to improve the therapeutic efficacy in the treatment of LPS-induced lung ALI and showed potential prospects in clinics.
7) Figure 11. In vivo biosafety validation of different formulations of Nar. (A) Time-axis of biosafety 603 determination: The mice were i.t administrated with different formulations of Nar (10mg/kg) once 604 a day for 7 days. Bronchoalveolar Fluid (BALF), blood, and lung tissues were collected at 7th day. 605 (B) H&E staining images of the major organs (lung, liver, spleen, kidney and heart) collected from 606 the mice after i.t. administration of Nar. Please, improve the description of the figure.
Response: Thanks a lot for this comment. We have revised the descriptions accordingly (shown in red words in Page 17, Page 34).

Round 2
Reviewer 1 Report
The authors improved the manuscript according to the suggestions.
The authors improved the manuscript according to the suggestions.
Reviewer 2 Report
Current paper seems relevant to its field. It's been reviewed and now it's ready to publish for me.
Authors replied to my comments in a satisfactorily way. IMHO this paper can now be published.
Reviewer 3 Report
I have no further comments. The responses from the authors and the revised manuscript are clearer.
I have no further comments. The manuscript has been improved as requested.